# Addressing Leakage in Concept Bottleneck Models

**Marton Havasi**
School of Engineering and Applied Sciences
Harvard University
mhavasi@seas.harvard.edu

**Sonali Parbhoo**
Department of Electrical Engineering
Imperial College London
s.parbhoo@imperial.ac.uk

**Finale Doshi-Velez**
School of Engineering and Applied Sciences
Harvard University
finale@seas.harvard.edu

## Abstract

Concept bottleneck models (CBMs) enhance the interpretability of their predictions by first predicting high-level concepts given features, and subsequently predicting outcomes on the basis of these concepts. Recently, it was demonstrated that training the label predictor directly on the probabilities produced by the concept predictor as opposed to the ground-truth concepts, improves label predictions. However, this results in corruptions in the concept predictions that impact the concept accuracy as well as our ability to intervene on the concepts – a key proposed benefit of CBMs. In this work, we investigate and address two issues with CBMs that cause this disparity in performance: having an insufficient concept set and using inexpressive concept predictor. With our modifications, CBMs become competitive in terms of predictive performance, with models that otherwise leak unintended information in the concept probabilities, while having dramatically increased concept accuracy and intervention accuracy.

## 1 Introduction

Building interpretable machine learning models is important in several high-stake domains where humans must typically interact with the models to make meaningful decisions. Among such models, Concept bottleneck models (CBMs) (Chen et al., 2020; Koh et al., 2020; Yeh et al., 2020; Lage & Doshi-Velez, 2020; Wang et al., 2017) propose explicitly aligning the intermediate layers of a neural network with some pre-defined expert concepts by first training a predictor of concepts, and subsequently using this to predict the label (Figure 1 left). CBMs are examples of interpretable models that are well-suited for human-AI joint decision making. A human supervisor can interpret and understand the label predictions by inspecting the concept predictions. Moreover, when the supervisor notices a misprediction among the concepts, they may intervene by correcting the concept prediction and, in turn, the model is able to update its final prediction accounting for the revised concept.

Typically when training a CBM, the concept predictor and the label predictor are trained independently on ground-truth concepts. Unfortunately, these models can underperform in predictive performance compared to a black-box predictor. Instead, the most predictive CBMs use soft concepts, meaning that the inputs to the label predictor are the concept probabilities (the numerical values between 0 and 1) as predicted by the concept predictor (Koh et al., 2020). Soft concepts rely on this information-rich representation, which is in contrast with hard concepts, where label predictor only accepts binary concepts.

36th Conference on Neural Information Processing Systems (NeurIPS 2022).

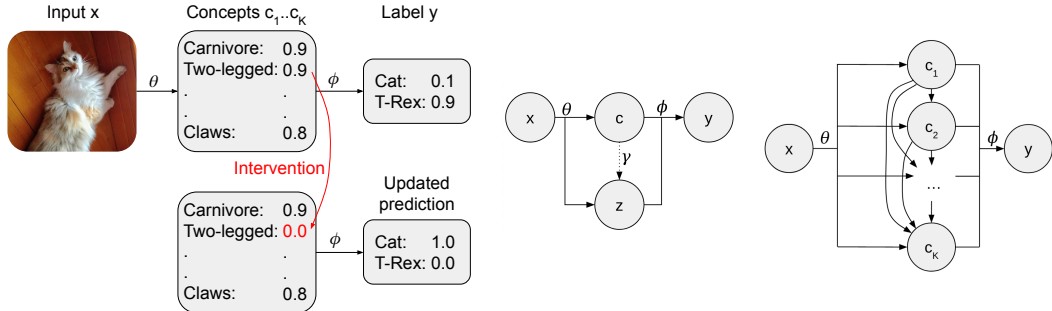

Figure 1: **Left:** The concept bottleneck model (CBM) makes its prediction in two stages. First, the concept predictor (parametrized by $\theta$) estimates the probability of each binary concept $c_1 \ldots c_K$ in the input $x$. Then, the label predictor (parametrized by $\phi$) predicts the label $y$. When the model mispredicts a concept, one can intervene by setting the concept probability to $0$ or $1$ and obtain an updated prediction. **Middle:** In a side-channel CBM, we model a set of latent concepts $z$ that are inferred from data. The concept predictor predicts both $c$ and $z$ ($p_\theta(c, z|x)$), and the label predictor predicts the label ($p_\phi(y|c, z)$). An amortization network is trained to rapidly infer $z$ from $c$ ($p_\gamma(z|c)$) when needed. **Right:** In an autoregressive concept predictor, concept $c_k$ is predicted given the input $x$, and all previous concepts $c_1 \ldots c_{k-1}$ ($p_\theta(c_k|x, c_{1:k-1})$). This enables the concept predictor to model correlations in the concepts.

While soft concepts may improve predictive performance, this improvement comes at a cost. The issue is that soft concepts allow the concept predictor to convey unintended information about the label that would otherwise not be available to the label predictor (Mahinpei et al., 2021; Margeloiu et al., 2021). For example, in an animal classifier, when the concept predictor recognizes an image as a cat, it may predict a slightly higher likelihood to the concept 'fur' than the concept 'tail', and when it recognizes the image as a dog, it may assign higher likelihood to the concept 'tail' than the concept 'fur'. Even though cats and dogs are indistinguishable based on these two concepts alone, the label predictor may still compare the two soft concept probabilities and make an accurate classification. A flexible predictor, such as a deep neural network, can quickly learn such encoding patterns and improve its prediction using the information leaked by the concept predictor. Thus, the problem is dubbed leakage.

Leakage compromises the interpretability and trustworthiness of the model. The concept predictor no longer needs to faithfully predict the concepts to obtain high-quality label predictions. Instead, it only needs to encode the class label in the soft concept probabilities. A human supervisor can never be sure if the model is predicting a certain concept because it is likely to be present, or because it is encoding for something else. Moreover, since interventions remove this extra information from the concept predictions, predictive performance conditional on interventions suffers significantly.

How can one avoid leakage? While hard concepts ensure that information is not leaked, these models suffer from poor performance in comparison to their soft counterparts. In this work, we focus on the performance disparity between soft and hard CBMs. Specifically, we attribute this disparity to (i) having an insufficient concept set and (ii) having an inflexible concept predictor. The former issue occurs as a result of the Markovian assumption under which CBMs operate: all the information about the label in the input must be captured by the concepts. If this assumption is not met, the performance of the hard CBM is limited by the information available in the concepts. The latter issue stems from the fact that CBMs predict concepts *independently* and cannot capture correlations between concepts. These correlations are important since some combinations of concepts may be unlikely or impossible (eg. the concepts 'fur' and 'scales' are mutually exclusive). These two issues hinder the performance of hard CBMs, while soft CBMs are able to mitigate them with leakage, at the cost of interpretability and intervenability.

We introduce two technical contributions that allow hard CBMs to match the predictive performance of soft CBMs without compromising their resilience to leakage. First, we circumvent the Markovian assumption by introducing a side-channel to learn a set of latent concepts alongside the known concepts that are needed for accurate predictions. Second, we introduce an autoregressive configuration

for the concept predictor that significantly improves their flexibility and it allows them to capture correlations in the concepts.

On several synthetic and real applications, we demonstrate that our side-channel models improve the performance of hard CBMs in terms of accuracy, without affecting concept predictions or intervenability. In addition, our autoregressive concept predictor significantly improves both concept accuracy and intervention accuracy compared to soft CBMs and non-autoregressive hard CBMs.

Summary of contributions:

- In our experiments, we demonstrate that leakage is detrimental to interpretability and intervenability in soft CBMs.

- Hard CBMs are resilient to leakage, but their performance is hindered by not meeting the Markovian assumption and having an inflexible concept predictor. We address these shortcomings:

  - We circumvent the Markovian assumption by introducing a side-channel to learn a set of latent concepts.
  - We increase the flexibility of the concept predictor by adapting an autoregressive architecture.

## 2  Background: Concept Bottleneck Models

Consider a dataset $\mathcal{D} = \{x^{(i)}, c^{(i)}, y^{(i)}\}_{i=1}^N$ consisting of $N$ triplets of the form: input $x \in \mathbb{R}^d$, concept vector $c \in \{0,1\}^K$ of $K$ binary concepts, and class label $y \in \mathbb{N}$ (we consider classification problems, but our results extend to regression problems where $y \in \mathbb{R}$). Formally, a CBM is comprised of a concept predictor, parametrized by $\theta$, which is trained to predict the concept vector from the input $p_\theta(c|x)$, and a label predictor, parametrized by $\phi$, subsequently trained to predict the label from the concepts $p_\phi(y|c)$ (Koh et al., 2020) (Figure 1 left).

**Interpretability and Interventions**  There are two key benefits to CBMs that make them well-suited for human-AI joint decision making. First, they explain their predictions by showing which high-level concepts the concept predictor was able to find in the data. Second, a human operator may intervene on these concepts to see how changing the concept values affects the final prediction. This is done by setting the probabilities of the concepts to $1$ or $0$ to indicate their presence or absence. The label predictor is able to adjust its prediction to account for the intervention (Figure 1 left).

**Hard CBMs**  In hard CBMs, the label predictor only accepts binary concepts as inputs. The concept predictor and the label predictor are trained independently on the ground truth data: $\hat{\theta} = \arg\max_\theta \mathbb{E}_\mathcal{D}[\log p_\theta(c|x)]$ and $\hat{\phi} = \arg\max_\phi \mathbb{E}_\mathcal{D}[\log p_\phi(y|c)]$ (here, we omit regularizers for brevity). To make predictions on a new input point, one must marginalize out the possible concepts: $p_{\theta\phi}(y|x) = \mathbb{E}_{c \sim p_\theta(c|x)}[p_\phi(y|c)]$.

**Soft CBMs**  Soft CBMs, on the other hand, pass the concept probabilities (the value in $[0,1]$) directly to the label predictor. When encountering a new input $x$, a soft concept model first calculates the output of concept predictor $g(x)$ where $\sigma(g(x)_k) = p_\theta(c_k = 1|x)$ for $k = 1 \dots K$. These probabilities are then forwarded to the label predictor, whose output is $f(c) = p_\phi(y|c)$. Combined, they produce the final prediction $f(g(x)) = p_{\theta\phi}(y|x)$.

Three different training strategies have been proposed to train soft CBMs (Koh et al., 2020). First, *independent training*, where the concept predictor and the label predictor are trained independently $\hat{\theta} = \arg\max_\theta \mathbb{E}_\mathcal{D}[\log p_\theta(c|x)]$ and $\hat{\phi} = \arg\max_\phi \mathbb{E}_\mathcal{D}[\log p_\phi(y|c)]$. Second, *sequential training*, where the label predictor is trained after the concept predictor $\hat{\theta} = \arg\max_\theta \mathbb{E}_\mathcal{D}[\log p_\theta(c|x)]$ and $\hat{\phi} = \arg\max_\phi \mathbb{E}_\mathcal{D}[\log p_{\theta\phi}(y|x)]$. Finally, *joint training*, where the concept predictor and the label predictor are trained jointly $\hat{\theta}, \hat{\phi} = \arg\max_{\theta\phi} \mathbb{E}_\mathcal{D}[\lambda \log p_\theta(c|x) + \log p_{\theta\phi}(y|x)]$. Here, $0 \leq \lambda$ is a scalar hyperparameter controlling the tradeoff between the two terms. High $\lambda$ prioritizes accurate concept predictions, while $\lambda$ close to $0$ focuses on accurate label predictions.

**Leakage** Leakage occurs in soft CBMs when trained sequentially or jointly (Mahinpei et al., 2021). The label predictor learns to utilize the additional, unintended information in the soft concept probabilities output by the concept predictor.

In the animal classification example, when trained jointly, the concept predictor can learn to assign slightly higher likelihood to the concept 'fur' than the concept 'tail' when it recognizes the image as a cat and predict higher likelihood to the concept 'tail' than the concept 'fur' when it recognizes the image as a dog. Despite that cats and dogs are indistinguishable based on these two concepts, the label predictor can compare the two soft concept probabilities and make an accurate classification. The label predictor is no longer limited by set of the pre-defined concepts: the concept predictor is able leak further information about the label to improve the predictive performance. This decreases the overall loss: the concept log-likelihood $\log p_\theta(c|x)$ decreases slightly due to the slight perturbation, but the label log-likelihood $\log p_{\theta\phi}(y|x)$ increases significantly since the label predictor now has access to the leaked information. This is further exacerbated by using $\lambda < 1$, which prioritizes the label log-likelihood over the concept log-likelihood in the training objective.

Leakage damages both interpretability and intervenability. Interpretability suffers because the explanation of the label is based not only on the concepts, but also on the behavior of the *specific* concept predictor. The human operator would need an in-depth knowledge of the concept predictor to understand why a small change in the likelihood of 'fur' leads to a significant change in the classification. Interventions are also adversely affected. When one intervenes on the 'fur' concept by setting its probability to 1, it can decrease the predictive accuracy because this removes all information encoded in the soft probability value.

Hard CBMs avoid leakage by only passing binary concepts to the label predictor. The concept predictor and the label predictor learn to faithfully predict the concepts from the inputs and the labels from the concepts respectively.

## 3   Related works

**Concept Activation Vector and Whitening Methods** Several methods exist to interpret the layers of a neural network and what they learn. Among these, Kim et al. (2018); Zhou et al. (2018); Ghorbani et al. (2019) use concept activation vector methods to analyse networks that have been pre-trained. The concept activation vectors are chosen to align with pre-defined or automatically-discovered concepts. These vectors are based on the assumption that the latent space can be analysed posthoc in a specific form. Other methods such as concept whitening (Chen et al., 2020) directly alter a given layer of a network by adding a module that aligns the latent space to certain axes of interest to understand the computations leading to that layer. Unlike our work, none of the former methods address the fundamental issue of leakage in CBMs that significantly impacts the interpretability and intervenability of these models.

**Soft Concept Bottleneck Models** Soft CBMs (Chen et al., 2020; Koh et al., 2020) propose passing *concept probabilities* (value in $[0, 1]$) to the label predictor to significantly improve prediction accuracy in comparison to CBMs that learn hard concepts. Though performant, several works show that slightly distorting the predictive concept probabilities enables the concept predictor to 'leak' the final label to the label predictor (Mahinpei et al., 2021; Margeloiu et al., 2021). We propose tools that allow us to diagnose and address these issues to directly improve the performance of hard CBMs, thereby overcoming the issue of information leakage in their soft counterparts.

**Information Bottleneck Methods** IB methods take a joint distribution $p(X, Y)$ and compress the inputs $X$ into a latent representation $Z$ that is most informative about the target $Y$ (Tishby et al., 2000). This latent representation may be thought of as a type of concept. Several methods have been proposed to use such a concept representation $Z$ to compute the effect of an intervention (Parbhoo et al., 2020; Chicharro et al., 2020). Both the traditional IB and the deep variational IB formulations assume specific Markov structures for the generative model (Alemi et al., 2016; Wieczorek & Roth, 2020). In contrast, our work uses side-channels to learn informative concepts that are predictive of the target while compressing the inputs, and is independent of such Markovian assumptions.

**Causal Concept Effect Models** Goyal et al. (2019); Feder et al. (2021) explain coupling a CBM with a causal model and computing the Causal Concept Effect to speculate about what might happen

to the prediction had a concept been different. Unlike this work, we examine only the statistical setting where we can intervene to correct or modify an incorrect concept prediction, but not to speculate about counterfactuals since the model does not capture causal relationships.

**Concept completeness** Yeh et al. (2020) study when a set of concepts is sufficient for explaining the prediction in a deep neural network. They propose an accuracy-based completeness score. Their score, however, does not imply that the concepts capture all information in the input i.e. the Markovian assumption. Our information-based completeness score indicates precisely when the assumption is met.

# 4 Addressing Two Shortcomings of Concept Bottleneck Models

In this section, we explain and address the performance disparity between soft CBMs and hard CBMs. We present two sources of information that CBMs are unable to capture: information that is not available in the concepts and information in the correlations in the concepts. Soft CBMs can lessen the impact through leakage, but hard CBMs cannot. We propose modifications to hard CBMs that allow the use of information from these two sources while retaining their resilience to leakage.

## 4.1 The Markovian Assumption

First, we consider the case when the concepts are insufficient for predicting the final label, but $x$ has more information that can help. We refer to this as the lack of a Markovian assumption. When the Markovian assumption holds, the chain $x \to c \to y$ is Markovian, which is defined as $y$ being conditionally independent of $x$ given $c$: $\mathrm{I}(y; x|c) = 0$, where I denotes the mutual information.

When the assumption holds, $c$ must contain all information about $y$ that is present in $x$. Otherwise, $c$ is insufficient for predicting $y$; therefore, a CBM that relies on hard concepts will underperform compared to a black box $x \to y$ predictor. In what follows, we describe how CBMs can still be applied with the help of a side-channel and define a concept completeness score to diagnose when the Markovian assumption may not hold.

### 4.1.1 Using a Side-channel

Along with the known concepts $c$, we model $L$ unknown concepts $z \in \{0,1\}^L$ in a side-channel CBM (Figure 1 middle). This setup allows the model to capture information about $y$ present in $x$ but not present in $c$ via the latent concepts.

Training the side-channel model is analogous to training a hard CBM. We optimize the predictive log-likelihood using stochastic gradient ascent:

$$\hat{\theta}, \hat{\phi} = \arg\max_{\theta, \phi} \mathbb{E}_{\mathcal{D}} \left[ \log p_\theta(c|x) + \log \mathbb{E}_{p_\theta(z|x)} \left[ p_\phi(y|c, z) \right] \right]. \tag{1}$$

A caveat is that the gradient of $\mathbb{E}_{p_\theta(z|x)} \left[ p_\phi(y|c, z) \right]$ need to be estimated using a gradient estimator. We use the reinforce estimator (Williams, 1992) because it is able to give an unbiased estimate of the gradients, and it works well in a low dimensional settings such as $z$:

$$\nabla_\theta \mathbb{E}_{p_\theta(z|x)} \left[ p_\phi(y|c, z) \right] = \mathbb{E}_{p_\theta(z|x)} \left[ p_\phi(y|c, z) \nabla_\theta \log p_\theta(z|x) \right]. \tag{2}$$

The gradient is then estimated using $M$ Monte-Carlo samples:

$$\nabla_\theta \mathbb{E}_{p_\theta(z|x)} \left[ p_\phi(y|c, z) \right] \approx \frac{1}{M} \sum_{z^{(m)} \sim p_\theta(z|x)} p_\phi(y|c, z^{(m)}) \nabla_\theta \log p_\theta(z^{(m)}|x) \tag{3}$$

At prediction time, we can decide whether to use the side-channel for prediction or rely purely on the known concepts. If the side-channel is used, the predictions are calculated using $M$ Monte-Carlo samples:

$$p_{\theta\phi}(y|x) \approx \frac{1}{M} \sum_{\substack{c^{(m)}, z^{(m)} \\ \sim p_\theta(c, z|x)}} p_\phi(y|c^{(m)}, z^{(m)}). \tag{4}$$

In applications where the label prediction must be fully explained by the concepts, we have to marginalize out the side-channel. This requires estimating $p_\theta(z|c)$. One can obtain an estimate using $p_\theta(z|c) \approx \mathbb{E}_{c',x' \sim \mathcal{D} \& c'=c}[p_\theta(z|x')]$; however, this requires iterating over the whole dataset each time a prediction is made.

Our solution is to train a small (1 hidden layer) predictor that can rapidly estimate $p_\theta(z|c)$ from $c$. We call this the amortization network, since it amortizes the cost of inferring $z$ from $c$. We denote its parameters $\gamma$, and train it using maximum-likelihood training: $\hat{\gamma} = \arg\max_\gamma \mathbb{E}_\mathcal{D}[\log p_\gamma(z|c)]$. In our experiments, the amortization network is trained over 10-20 epochs depending on the dataset. Equipped with $\gamma$, we can make predictions without using the side-channel:

$$\mathbb{E}_{p_\theta(c|x)}[p_\phi(y|c)] \approx \frac{1}{M} \sum_{\substack{c^{(m)} \sim p_\theta(c|x) \\ z^{(m)} \sim p_\gamma(z|c^{(m)})}} p_\phi(y|c^{(m)}, z^{(m)}). \tag{5}$$

A second benefit marginalizing the side-channel with the amortization network is that it can help us understand how much the model relies on the side-channel for prediction. To formalize this concept, we introduce a completeness score.

**A Completeness Score** The side-channel model can be used to estimate what fraction of the relevant information is present in $c$. This is useful for diagnosing when a side-channel is needed and to understand how much the model relies on it. If the model heavily relies on the side channel, it indicates that key concepts are missing from $c$. We define

$$\mathcal{C} = \frac{\mathrm{I}(y;c)}{\mathrm{I}(y;c,x)}. \tag{6}$$

$0 \leq \mathcal{C} \leq 1$, because its a ratio of non-negative quantities and the nominator is less-than-or-equal to the denominator since $\mathrm{I}(y;c,x) - \mathrm{I}(y;c) = \mathrm{I}(y;x|c) \geq 0$ (Cover, 1999). When $\mathcal{C} = 1$, we have $\mathrm{I}(y;c) = \mathrm{I}(y;c,x)$, so $\mathrm{I}(y;c,x) - \mathrm{I}(y;c) = \mathrm{I}(y;x|c) = 0$ thus the Markovian assumption holds, and $c$ captures all relevant information. When $\mathcal{C} = 0$ ($\mathrm{I}(y;c) = 0$ i.e. $c$ and $y$ are independent), none of the information in $c$ is relevant for predicting $y$. With the help of the amortization network, it is possible to estimate $\mathcal{C}$:

$$\mathcal{C} \approx \frac{\mathrm{H}(y) + \mathbb{E}_\mathcal{D}\left[\log \mathbb{E}_{z \sim p_\gamma(z|c)}[p_\phi(y|c,z)]\right]}{\mathrm{H}(y) + \mathbb{E}_\mathcal{D}\left[\log \mathbb{E}_{z \sim p_\theta(z|x)}[p_\phi(y|c,z)]\right]}. \tag{7}$$

**Interpretability of the Side-channel** The latent concepts captured by the side channel can be inspected for interpretability. They may align with human understandable concepts (in which case, they can be added to $c$), but it is possible that they are uninterpretable. The side-channel model should still be used to avoid the detrimental effects of leakage on concept accuracy and interventions. Note that we have agency on deciding whether to use the side channel information at prediction time or not. Including the side-channel information is ideal for applications where accuracy is high-priority and partial explanations are sufficient (Equation 4). For applications where the human operator must know the full set of concepts contributing to the prediction, we can make predictions without it (by marginalizing $z$ with the help of the amortization network as shown in Equation 5).

### 4.2 The Expressivity of the Concept Predictor

The second source of information that CBMs are unable to capture is correlations in the concepts. An issue with the usual formulation of a CBM is that the concept predictor predicts the concepts independently (conditional on the input). This may give poor performance when the concepts are correlated. Consider two concepts that are mutually exclusive. A predictor that predicts them independently does not have the flexibility to express their mutual exclusivity and it inadvertently predicts non-zero probability of both concepts being present. While a soft CBM can learn during training that the concept probabilities correspond to mutually exclusive concepts, a hard CBM cannot.

We propose a modification to the CBM architecture that allows hard CBMs to capture correlations in the concepts. Our solution is to use an autoregressive architecture for concept prediction similarly to classifier chains in multilabel classification (Read et al., 2011; Dembczynski et al., 2010). To predict

the $k$-th concept, use not only the input $x$, but also the already predicted concepts $c_{1:k-1}$ [1] (Figure 1 right).

The log-likelihood, which is needed for training, can be computed by iterating over the concepts: $\log p_\theta(c|x) = \sum_{k=1}^{K} \log p_\theta(c_k|x, c_{1:k-1})$. To sample a concept set, which is needed for prediction, we must sample the concepts iteratively: $c_k \sim p_\theta(c_k|x, c_{1:k-1})$ for $k = 1 \ldots K$.

Lastly, interventions require a modified sampling algorithm. In autoregressive models, we cannot easily generate Monte-Carlo samples from the concept distribution conditional on a set of interventions. Instead, to estimate $p_{\theta,\phi}(\hat{y}|\hat{x}, \{\hat{c}_i\}_\mathcal{I})$ (where $\{\hat{c}_i\}_{i \in \mathcal{I} \subseteq \{1 \ldots K\}}$ denotes the set of intervened concepts), we propose a normalized importance sampling algorithm that generates $M$ weighted concept samples for prediction $p_{\theta,\phi}(\hat{y}|\hat{x}, \{\hat{c}_i\}_\mathcal{I}) \approx \frac{\sum_{m=1}^{M} w_m p_\phi(\hat{y}|c^{(m)})}{\sum_{m=1}^{M} w_m}$. Our proposal distribution $c^{(1)} \ldots c^{(M)} \sim q(c)$ matches the interventions $\{\hat{c}_i\}_\mathcal{I}$ when the concepts are intervened on and selects a random concept conditional on the previous concepts when they are not intervened on:

$$q(c_k) = \begin{cases} \delta(c_k, \hat{c}_k) & k \in \mathcal{I} \\ p_\theta(c_k|x, c_{1:k-1}) & k \notin \mathcal{I} \end{cases}, \tag{8}$$

where $\delta$ denotes Kronecker's delta function. To obtain an importance weighted estimate, we must compute the weight of each of the $M$ samples $w_m = \frac{p_\theta(c^{(m)}|\hat{x}, \{\hat{c}_i\})}{q(c^{(m)})}$. This is done by iterating over the $K$ concepts:

$$w_m = \prod_{k=1}^{K} \begin{cases} p_\theta(c_k^{(m)}|\hat{x}, c_{1:k-1}^{(m)}) & k \in \mathcal{I} \\ 1 & k \notin \mathcal{I} \end{cases}, \tag{9}$$

since the probability mass ratio $\frac{p_\theta(c_k^{(m)}|x, c_{1:k-1}^{(m)})}{q(c_k^{(m)}|x, c_{1:k-1}^{(m)})} = 1$ when $k \notin \mathcal{I}$. Finally, we obtain the importance weighted estimate $p_{\theta,\phi}(\hat{y}|\hat{x}, \{\hat{c}_i\}_\mathcal{I}) \approx \frac{\sum_{m=1}^{M} w_m p_\phi(\hat{y}|c^{(m)})}{\sum_{m=1}^{M} w_m}$. Note that due to this re-weighting scheme, interventions are able to affect concept predictions of earlier concepts. We also show this experimentally in Appendix E. Normalization is important to ensure the predictive distribution is proper. Psudocode for the interventions is shown in Appendix D.

## 5 Results and Discussion

We present empirical results showcasing the efficacy of our proposed modifications to concept bottleneck models (CBM).

### 5.1 Datasets and Models

We used two datasets in our experiment. First, an Early Warning Score (EWS) prediction task based on electronic health records data that simulates a potential use case. Second, to showcase the versatility of CBMs and compare to prior work, we apply our method to an image-based bird recognition task. Our code is available at `https://github.com/dtak/addressing-leakage`.

**MIMIC-III EWS (Johnson et al., 2016)** The task is to predict an EWS for patients at a hospital that informs the medical professionals about abnormalities in the vital signs that often precede deterioration and cardiac arrests (Subbe et al., 2001). The concepts ('Mild hypotension', 'Severe fever' etc.) provide justification for the score. The EWS ranges from 0 to 15, and it is based on the patient's vitals. Deviations from the norm increase the score depending on severity. For example, 'severe hypotension' adds +3 to the EWS, while 'mild fever' adds +1. The dataset does not contain concept annotations, so we defined them synthetically (Appendix A). The dataset contains records from 17,289 patients over a combined N=796,250 time steps (split into 530,802 training and 265,448 test examples, while ensuring that no patient appears in both sets). It has 13 input features, $K = 22$ concepts and 16 label classes (EWS from 0 to 15).

For MIMIC-III EWS, the concept predictor is a two-layer feed-forward neural network with a hidden layer of size 100, and the label predictor is a two-layer network with hidden layer of size 50. In the

---

[1] $c_{1:k-1}$ is a shorthand notation for $c_1 \ldots c_{k-1}$

Table 1: Concept and label accuracies (%) in soft and hard CBMs. The best result is bolded in each column. Hard autoregressive models (Hard AR) excel at concept accuracy and they are on-par with soft joint models in label accuracy.

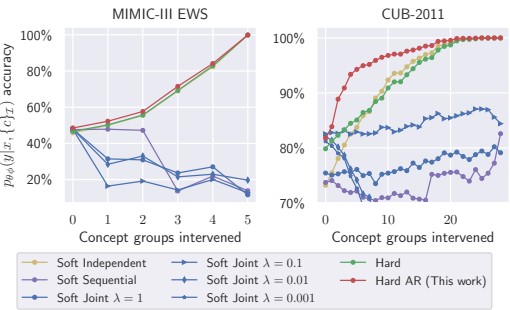

| | MIMIC-III EWS | | CUB-2011 | |
| Model | $p_\theta(c\|x)$ | $p_{\theta\phi}(y\|x)$ | $p_\theta(c\|x)$ | $p_{\theta\phi}(y\|x)$ |
|---|---|---|---|---|
| Soft Independent | 38.7±0.0 | 45.9±0.1 | 48.1±0.8 | 72.8±0.8 |
| Soft Sequential | 38.7±0.0 | 47.6±0.0 | 46.1±0.4 | 73.9±0.1 |
| Soft Joint $\lambda = 1$ | 38.7±0.0 | 48.0±0.0 | 47.7±0.2 | 75.4±0.1 |
| Soft Joint $\lambda = 0.1$ | 35.6±0.1 | 48.0±0.0 | 47.9±0.4 | **82.7±0.2** |
| Soft Joint $\lambda = 0.01$ | 32.1±0.1 | 47.9±0.0 | 0.1±0.0 | 82.1±0.3 |
| Hard | 38.7±0.0 | 46.6±0.0 | 68.0±0.2 | 79.5±0.3 |
| Hard AR (This work) | **41.0±0.0** | **48.5±0.0** | **81.4±0.1** | 81.7±0.2 |

Figure 2: Label accuracy after intervening on subset of concepts. Soft joint CBMs deteriorate quickly, while hard CBMs, including our proposed autoregressive model (Hard AR), improve with interventions.

autoregressive case, a small, two-layer network (hidden layer size 20) predicts each concept. The input to these networks are the previously predicted concepts, concatenated with the 100 hidden units of the concept predictor. We use $M = 200$ Monte-Carlo samples for prediction. For training hyperparameters, see Appendix C.

**Caltech-UCSD Birds 2011 (Wah et al., 2011)**  For an accurate comparison to Koh et al. (2020), we replicated their results with the same preprocessing steps. This dataset contains N=11,788 (5,994 training and 5,794 test) images of 200 bird species native to North America, along with 312 binary attributes (eg.: 'wing-color' and 'beak-shape'). Following Koh et al. (2020), we narrow down this list of attributes to $K = 112$ binary concepts, only including attributes that are present for at least 10% of the dataset. To de-noise the attributes, we consolidate the concepts within each species by majority voting (where if the majority of ravens have black wings, we ensure that all ravens in the dataset are annotated as having black wings (Koh et al., 2020)).

For CUB-2011, we fine-tuned a pre-trained Inception v3 network (Szegedy et al., 2016) as the concept predictor. Following Koh et al. (2020), the label predictor for the soft models is a simple linear predictor, while for hard models, we use a two-layer neural network with a hidden layer of size 200. In the autoregressive case, a small, two-layer network (hidden layer size 50) predicts each concept. The input to these networks are the previously predicted concepts concatenated with the output of the Inception network. We use $M = 200$ Monte-Carlo samples for prediction. For training hyperparameters, see Appendix C.

### 5.2 Autoregressive Concept Predictions

Our first set of experiments quantitatively assess the impact of leakage as well as examine the effectiveness of autoregressive concept predictors. We lead with these experiments to justify the use of the autoregressive architecture in our subsequent side-channel experiments. It is important to note that both datasets have complete concept sets (they meet the Markovian assumption $\mathcal{C} = 1$) so any performance disparity is due to the inflexibility of the concept predictor.

**Leakage makes soft CBMs unsuitable for tasks where interpretability or intervenability are required.**  Soft CBMs underperform in concept accuracy, but they are often able to predict the label correctly (Table 1). This is only possible if the concepts are leaking information about the label in an uninterpretable way. Interventions, where the human operator sets concepts to their true values, are ineffective on sequential and joint soft CBMs (Figure 2). This should improve their predictions and yet they still mispredict the label even when all concepts are supplied, because interventions remove the leaked information from the soft concept probabilities.

**Autoregressive predictors close the gap between soft and hard CBMs in label accuracy while having dramatically increased concept accuracy.**  In Table 1, we see that soft CBMs perform well in label accuracy, but leakage is detrimental to their concept accuracy. The hard CBM performs better in concept accuracy, but its performance is limited by the inflexibility of the concept predictor. The autoregressive model (Hard AR, this work) offers the best concept accuracy with competitive

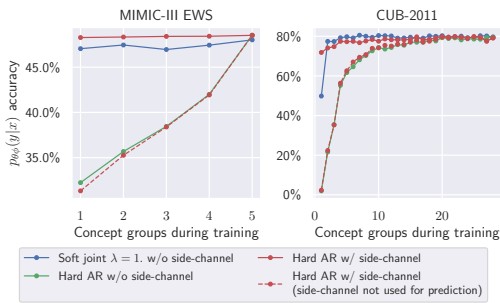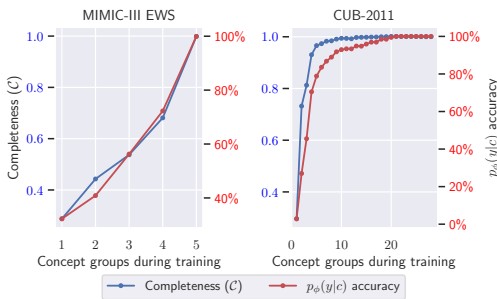

Figure 3: **Left:** The predictive performance of soft joint CBMs and hard CBMs. The side-channel explains the gap between the hard and soft models. It captures the information that is leaked in the soft joint CBM but not captured by the hard CBM. **Right:** The completeness score and label accuracy of a hard autoregressive CBM with a side-channel. The x-axis denotes the number of concept groups available during training. When the completeness score reaches 1, the Markovian assumption holds by definition.

label accuracy: in the case of MIMIC-III EWS, Hard AR is above soft models in both concept and label accuracy, while in CUB-2011, Hard AR has the best concept accuracy with label accuracy within 1% of joint models. (Note that concept accuracy refers to predicting all concepts correctly simultaneously as opposed to averaging the accuracies over the different concepts.)

**Hard autoregressive models excel at interventions while soft CBMs fail to adjust their predictions.** On Figure 2, we see that the performance of soft sequential and soft joint models deteriorate quickly due to leakage, while soft independent and hard models improve with interventions. The best performing model is the Hard AR (this work) on both datasets.

### 5.3   Using a Side-channel

Next, we examine the side-channel model as well as our proposed completeness score. In these experiments, we trained CBMs with only a subset of the available concepts to simulate the scenario where the Markovian assumption does not hold. We used $L = 20$ latent concepts for MIMIC-III and $L = 100$ latent concepts for CUB-2011 (we settled on these values because larger $L$ yielded no significant improvement in predictive performance). Note that we use autoregressive concept predictors in hard CBMs, so any performance disparity is due to having an insufficient concept set.

In MIMIC-III EWS, the concepts were grouped depending on which of the five vital signs they are derived from (blood-pressure, temperature, respiratory rate, heart rate, oxygen saturation). In CUB-2011, a group of concepts refer to related attributes, such as 'wing-color'.

**The side-channel closes the gap between soft and hard CBMs when the Markovian assumption is not met.** In Figure 3 (left), we see that the performance of the side-channel matches the predictive performance of the joint model when accounting for the latent concepts and it performs equivalently to the no side-channel model when the latent concepts are marginalized with the help of the amortization network. Therefore, the side-channel models precisely the information that is not present in the concepts, i.e. the information that is leaked in the soft joint model.

**The completeness score indicates when the Markovian assumption is fulfilled.** The completeness score proves to be a useful diagnostic tool to show when the available concept set limits the performance of hard CBMs. Figure 3 (right) depicts the completeness score $\mathcal{C}$ depending on the number of concepts used during training. As the concept set grows, $\mathcal{C}$ tends towards 1, and reaches it once the concept set contains all information about the final label, and the Markovian assumption is met. Interestingly, completeness reaches 1 before $p_\phi(y|c)$ accuracy peaks in CUB-2011. This is possible when a concept contains information to predict $y$, but this information is not present in $x$ (e.g. the color of the belly is not visible on images where it could be the deciding factor). Such concepts do not contribute to completeness, but they increase label accuracy when the concepts are given.

Table 2: The computation costs of different models. The times are recorded on a V100 GPU. These times should be considered approximate as we were unable to account for factors such as the utilization of the assigned compute node.

| Dataset | Model | Time/epoch | Training epochs |
|---|---|---|---|
| MIMIC-III EWS | Soft Independent/Sequential/Joint | 24.2 s | 20 |
| | Hard | 27.0 s | 20 |
| | Hard AR | 173.4 s | 20 |
| | Hard AR w/ side channel | 308.7 s | 30 |
| CUB-2011 | Soft Independent/Sequential/Joint | 52.7 s | 60 |
| | Hard | 51.0 s | 150 |
| | Hard AR | 83.2 s | 200 |
| | Hard AR w/ side channel | 83.7 s | 210 |

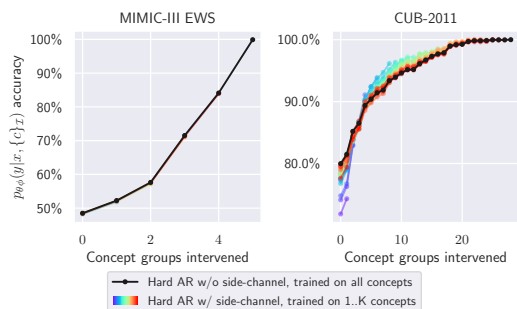

Figure 4: Interventions remain effective when trained with a side-channel on partial concepts. The color wheel denotes the number of concepts groups used during training: violet corresponds to a single concept group and red corresponds to almost all concept groups used. The performance is identical for all models in MIMIC-III EWS and the lines completely overlap.

**The side-channel is fully compatible with interventions** There have been concerns that a side-channel model could encode the label information in the side-channel, thus rendering interventions ineffective (Koh et al., 2020). We found that this is not the case. Figure 4 shows that interventions are as effective when trained with a subset of concepts as when trained with the complete concept set. In the case of MIMIC-III EWS, the intervention accuracy of the side-channel models completely overlap with the hard CBM trained on all concepts. In CUB-2011, the interventions are less effective with 1-2 concept groups, but they quickly catch up to the model trained with the complete concept set.

**Computational costs.** The benefits of autoregressive models and side-channel models come at an increased computational cost. Table 2 shows the computational cost of each model. Notably, the more complex image dataset, CUB-2011 takes more epochs to converge. The cost of the autoregressive architecture and the side-channel is more significant when the predictive model is relatively simple, which is the case in MIMIC-III EWS.

## 6 Conclusions

Leakage is detrimental to model interpretability and intervenability in soft concept bottleneck models. Hard CBMs are resilient to leakage, but they lag behind their soft counterparts in predictive performance. We exhibited and addressed two causes of performance disparity between soft and hard CBMs. Firstly, not meeting the Markovian assumption, which can be diagnosed using our proposed completeness score and addressed using a side-channel model. Secondly, the inflexibility of the concept predictor, which can be mitigated by adapting an autoregressive architecture.

With our modifications, CBMs no longer suffer from leakage and have significantly improved concept accuracy and intervention accuracy. This makes them more interpretable and more suitable for human-AI joint decision-making tasks.

In future work, we hope to explore methods to present the side-channel information in a human-interpretable way. This would allow us to discover concepts that are needed to complete the concept set for a given task.

## Acknowledgements

This work was funded by a grant from the National Institute of Mental Health (grant no. R01MH123804). The funders had no role in the design and conduct of the study; collection, management, analysis, and interpretation of the data; preparation, review, or approval of the manuscript; or decision to submit the manuscript for publication.

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
