# A MIMIC-III EWS

This early warning score (EWS) dataset is built on real patient data recorded in hospitals. It consists of vital signs (blood-pressure, hearth rate etc.) and static features (age, gender). Unfortunately the dataset has no readily available concepts that we could use for EWS prediction. Instead, we synthetically define concepts based on the recommendation of Subbe et al. (2001).

Our concepts are concerned with five vital signs: respiratory rate, spo2 saturation, temperature, blood pressure and heart rate, with three severity levels: mild, moderate and severe. The contribution of each concept depends on the severity level. Mild conditions contribute +1, moderate contribute +2 and severe contribute +3 points up to a maximum EWS score of 15.

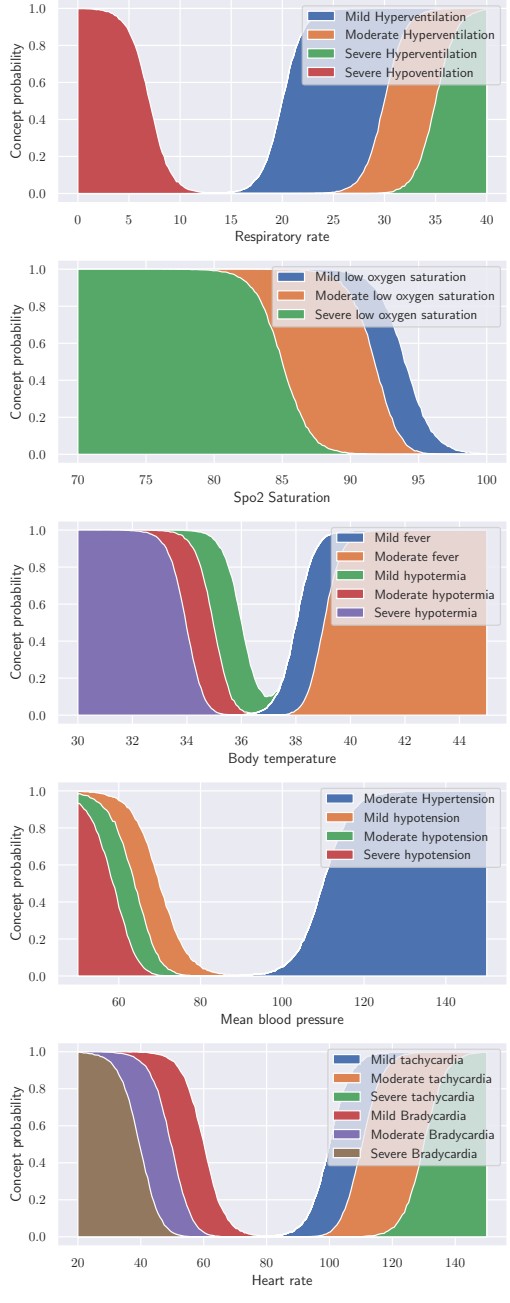

Figure 5: The probability of each concept depending on the vital signs in MIMIC-III EWS.

---

**Algorithm 1** Importance sampling for interventions in autoregressive models

---

**Input:** data $\hat{x}$, intervened concepts $\{\hat{c}_i\}_{i \in \mathcal{I} \subseteq \{1...K\}}$.
**for** $m = 1$ **to** $M$ **do**
   $w_m \leftarrow 1$
   **for** $k = 1$ **to** $K$ **do**
     **if** $k \in \mathcal{I}$ **then**
       $w_m \leftarrow w_m p_\theta(\hat{c}_k | x, c_{1:k-1}^{(m)})$
       $c_k^{(m)} \leftarrow \hat{c}_k$
     **else**
       $c_k^{(m)} \sim p_\theta(c_k | x, c_{1:k-1}^{(m)})$
     **end if**
   **end for**
**end for**
**Result:** $p_{\theta,\phi}(\hat{y} | \hat{x}, \{\hat{c}_i\}_{\mathcal{I}}) \approx \frac{\sum_{m=1}^{M} w_m p_\phi(\hat{y}|c^{(m)})}{\sum_{m=1}^{M} w_m}$

---

Figure 5 shows how we defined the probability of each concept based on the vitals. The transition between the severity levels is done using sigmoid curves. The reason for not using exact cutoffs is to synthetically simulate that the severity also depends on conditions not observed in the dataset. For example, the healthy range of for blood pressure depend on the physiology of patients such as weight, activity level etc.

## B  Note on interventions in sequential and joint models

For independently trained $g$ and $f$, the concept probabilities are directly passed to the label predictor $f(\sigma(g(x)))$, while for sequentially and jointly trained models, the logits of the concept probabilities are passed $f(g(x))$.

During interventions, the logits of probabilities 0 or 1 would correspond to $\pm\infty$. The network cannot handle infinite values, therefore we apply the modification proposed in Koh et al. (2020): when intervening on sequential or joint models, if the intervention indicates the absence of the concept, the concept likelihood is set to the 5th percentile concept likelihoods of negative examples in the training distribution, while if it indicates the presence of a concept, the likelihood is set to the 95th percentile of positive examples (Koh et al., 2020).

## C  Training hyperparameters

In MIMIC-III EWS, the models are trained over 20 epochs with SGD with batch size 512 and learning rate 0.001 decaying at the rate of 0.1 every 6.7 epochs. The amortization network is trained over 10 epochs with Adam (Kingma & Ba, 2014) with learning rate 0.001.

In CUB-2011, for the soft models, we replicated the results of Koh et al. (2020): we train over 60 epochs with SGD with batch size 64, weight decay 0.0004 and learning rate 0.01 decaying at the rate of 0.1 every 20 epochs.

Our autoregressive models (no side-channel) are trained over 150 epochs with SGD with batch size 64, weight decay 0.0008 and learning rate 0.03 decaying at the rate of 0.1 every 50 epochs. We also pre-train the autoregressive predictors (while keeping the inception network frozen) over 50 epochs with Adam (Kingma & Ba, 2014) with learning rate 0.03. The amortization network is trained over 20 epochs with Adam with learning rate 0.001.

The side-channel models are trained over 100 epochs with SGD with batch size 128, weight decay 0.0001 and learning rate 0.01 decaying at the rate of 0.1 every 33 epochs. The amortization network is trained over 20 epochs with Adam with learning rate 0.001.

## D  Autoregressive interventions

This section presents the pseudocode for making predictions with interventions in autoregressive models to show how a the practical implementation works (Algorithm 1). The algorithm itself is described in Section 4.2.

## E   Evaluating interventions

This section shows empirically that interventions are effective regardless of the ordering of the concepts due to the sample re-weighting scheme introduced in Section 4.2. The results are shown in Figure 6. There is a slight discrepancy between the original and the reverse order models at 10 concept groups. We experimented with increasing the number of MC samples from $M = 200$ to $M = 1000$ but the discrepancy remained (Figure not included here), which suggests that former order may be easier for the model to learn (with the given hyperparameters). Future work may consider investigating which concept orders are the easiest for the model to learn.

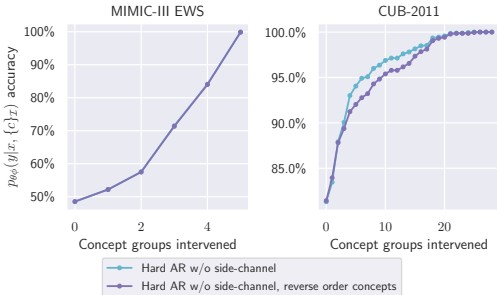

Figure 6: The reverse model is trained with the concepts in reverse order, meaning that we first intervene on the concepts that are last in the autoregressive architecture. The interventions remain effective due to our re-weighting scheme.