# OpenReview forum: "Addressing Leakage in Concept Bottleneck Models"
_NeurIPS.cc/2022/Conference — NeurIPS 2022 Accept_

### Official Review · Reviewer_YNur · 2022-07-07

**Rating:** 7
**Confidence:** 4
**Soundness:** 3 good
**Presentation:** 3 good
**Contribution:** 3 good

**Summary:**

The authors present a novel concept bottleneck model (CBM) aimed at
retaining the interpretability and suitability to be combined with
human interventions of hard CBMs (as compared to soft CBM), while
addressing two major limitations of hard CBM that they identify as
potential causes affecting their predictive accuracy: the inability to
deal with correlated concepts, and the problem of having an incomplete
concept set.

**Questions:**

How crucial is the marginalization on concepts for the performance in
hard CBM? how would an approach maximizing on concepts rather than
marginalizing perform?

**Limitations:**

The authors highlight the increased training cost due to the
autoregressive and side-channel components, which is however not
dramatic. One issue that would be worth discussing is the impact on
interpretability of the marginalization on concepts that is performed
on hard CBM, and how the marginalization on side concepts further affects it.

**Strengths And Weaknesses:**

The work is definitely relevant, given the importance that
interpretable neural networks and concept-based models have in
boosting trustworthiness and reliability of deep networks. The
proposed architecture is novel and all contributions are appropriately
motivated, with each contribution addressing one of the shortcomings
of existing hard CBMs.

The paper is sound in terms of justifications for the different
components introduced (the side channel, the autoregressive model) and
the strategies used for training, prediction and for dealing with
intervention, as well as for the introduction of the information
theoretical completeness score.

---

> ### Author Response · Authors · 2022-08-02
> **We appreciate the positive feedback.**
>
> Thank you for reviewing our work. We appreciate the positive feedback. We address each point in our reply.
>
> * **How crucial is the marginalization on concepts for the performance in hard CBM? how would an approach maximizing on concepts rather than marginalizing perform?**
> The $p(y|x)$ uncertainty estimates would suffer significantly. Without marginalization, the label predictor would be overconfident in predicting $y$ when presented with $c$, since it cannot account for the uncertainty in $c$.
>
> * **One issue that would be worth discussing is the impact on interpretability of the marginalization on concepts that is performed on hard CBM, and how the marginalization on side concepts further affects it.**
> In [Koh et al., 2020], the concept probabilities and label probabilities are communicated as percentages and the label prediction accounts for the uncertainty in the concepts. We expect our model to work similarly. A key difference is that each time a concept is intervened on (set to 100% or 0%), all the other concept probabilities and the label prediction need to be updated to account for the intervention (soft CBMs do not model concept correlations so they only need to update the label prediction after intervention). Our approach is preferred as it reflects that the concept probabilities change when provided new evidence.
> Regarding the side-channel, the completeness score we present is helpful to indicate how much the model uses on the side-channel versus the concepts for prediction. It helps  indicate when the model heavily relies on the possibly uninterpretable side-channel for prediction.

---

> > ### Comment · Reviewer_YNur · 2022-08-08
> > **Thanks for your clarifications**
> >
> > Thanks for clarifying these aspects, I encourage you to add a brief discussion to the paper, especially for the second point. These are however minor points, and my evaluation of the paper is definitely positive.

---

### Official Review · Reviewer_skST · 2022-07-09

**Rating:** 6
**Confidence:** 4
**Soundness:** 3 good
**Presentation:** 2 fair
**Contribution:** 3 good

**Summary:**

This paper considers the problem of leakage for concept bottleneck models (CBM). CBMs are models that predict concepts from features and then predict labels from concepts. When concept probabilities contain information about the label, leakage happens, which decreases intervention performance: when users intervene by changing concepts. Hard CBMs use binary concepts and avoid leakage but have lower performance. This paper proposes two methods for improving the performance of hard CBMs: (1) autoregressive where a concept is predicted using features and all previous concepts, (2) side channel where the label is predicted from concepts and directly from features. In experiments, the authors found that (1) autoregressive improves performance, especially intervention performance when a few concepts are changed, (2) side-channel improves performance when a few concepts are available.


**Questions:**

See my suggestions in the weaknesses section above.

**Limitations:**

The authors have adequately addressed the limitations.

**Strengths And Weaknesses:**

Strengths:
- CBMs are important models that support intervention. Leakage is a major issue with CBMs.
- Experimental results are mostly positive.

Weaknesses:
- Clarity: while I understand the paper now, I did get confused on my first read. The many ideas being presented got a bit mixed up. I think a reorganization will improve the clarity of the paper.

- Novelty: The application of autoregressive and side channeling to CBMs is novel, but these are fairly common modeling techniques, so the novelty is not high.

- Experiment 1 showed that autoregressive hard CBMs have better intervention performance when a small number of concepts (e.g. 5 out of 20) are changed. I think this was because the changed concepts propagate to other concepts, so it only works well for changing concepts that are around the beginning. This may also make it more difficult for users: changing one concept makes other concepts change. I think these should be tested in experiments and discussed.

- Experiment 2 showed that side channeling improved hard CBM when a small number of concepts are available. With a side channel, the model can predict the label directly from features, so this improvement is not surprising. The original CBM paper discussed side channeling as a potential future work, but pointed out a drawback: "cannot cleanly intervene on a single concept". So I think this means the intervention performance will be lower. This should also be tested in experiments.

---

> ### Author Response · Authors · 2022-08-02
> **Thank you for the constructive feedback.**
>
> Thank you for the constructive feedback on how to improve the paper. We address each point in our reply.
>
> * **Clarity: while I understand the paper now, I did get confused on my first read. The many ideas being presented got a bit mixed up. I think a reorganization will improve the clarity of the paper.**
> We are sorry that the first read was confusing. We close the performance gap between hard and soft CBMs.  Our two key contributions are: i) we address the inexpressivity of hard CBMs by using an autoregressive architecture and ii) we address the insufficiency of the concept set by using a side-channel. With the proposed changes, hard CBMs match the predictive performance of soft CBMs, while having dramatically increased concept and intervention accuracy.
> We are very open to suggestions to improve the presentation, so please let us know if there is a specific part that you recommend we change.
>
> * **Novelty: The application of autoregressive and side channeling to CBMs is novel, but these are fairly common modeling techniques, so the novelty is not high.**
> To our knowledge, we are the first ones to identify the insufficiency of the concepts and the inexpressivity of the concept predictor as two causes of the performance gap between soft and hard CBMs, and we are the first ones to apply these techniques with success in order to address the performance gap in CBMs.
>
> * **Experiment 1 showed that autoregressive hard CBMs have better intervention performance when a small number of concepts (e.g. 5 out of 20) are changed. I think this was because the changed concepts propagate to other concepts, so it only works well for changing concepts that are around the beginning.**
> Interventions are effective at any point in the concepts, regardless of ordering. In Equation 9, we show how the concept samples are re-weighted based on how well they predict the intervened concept, meaning that if $c_K$ is intervened on, it will affect the predictive probabilities of $c_1$ … $c_{K-1}$ through the sample weights $w_{1\dots M}$. Importantly, intervening on any concept thus affects the predictive probabilities of *all* other concepts.
> We also present empirical evidence for this. We compare against a model trained with the concept order reversed, meaning that we first intervene on the concepts that are last in the autoregressive architecture. The interventions remain effective due to our re-weighting scheme.
> Figure: https://imgur.com/a/FmO4NFK (Figure also available in Appendix E of the revised version)
>
> * **Experiment 2 showed that side channeling improved hard CBM when a small number of concepts are available. With a side channel, the model can predict the label directly from features, so this improvement is not surprising. The original CBM paper discussed side channeling as a potential future work, but pointed out a drawback: "cannot cleanly intervene on a single concept".**
> Indeed, we are aware of these concerns. We tested this claim (the results were included in Appendix D) and we found that intervention performance is near between having a large concept set with a side-channel and a small concept-set with a side channel. On MIMIC-III, the performance curves are overlapping, while on CUB-2011, we observe slightly lower intervention performance for the first 1-2 concept groups and close-to identical intervention performance afterwards.
> Figure: https://imgur.com/a/P00mIlK (Figure also available in Section 5.3 of the revised version)
>
> We are making the following changes to the revised version to address these concerns:
> 1. We are clarifying in Section 4.2 that interventions are effective regardless of the order of the concepts and include the experiments with reverse order concepts in Appendix E.
> 2. We are bringing the experiments from Appendix D into the main paper. This way, the concern about the efficacy of interventions when using a side-channel is addressed in a more prominent place.
>
> Let us know your thoughts on the revised version. If our reply addressed your concerns, please consider updating your score.

---

> > ### Comment · Reviewer_skST · 2022-08-08
> > **Thanks for replying**
> >
> > Thanks for answering my questions and addressing my concerns. I will increase my score.

---

> > > ### Author Response · Authors · 2022-08-09
> > > **Thank you**
> > >
> > > We appreciate the response, and we are glad that our reply addressed your concerns.
> > >
> > > Please remember to update your score before the metareviewer discussion begins by editing your original review.

---

### Official Review · Reviewer_L4TG · 2022-07-16

**Rating:** 6
**Confidence:** 4
**Soundness:** 2 fair
**Presentation:** 2 fair
**Contribution:** 3 good

**Summary:**

This work attends to the problem of “leakage” in the soft concept bottleneck models (CBMs): the soft concepts can convey unintentional information about the label, which compromises the interpretability and intervenability of a CBM -- the main purpose of such models -- even if it may improve the label prediction accuracy. The paper suggests two techniques: (1) a side channel to learn latent concepts to complement the insufficient (known) concepts under the Markovian assumption and (2) an autoregressive model to learn correlation among concepts. The authors conduct experiments on MIMIC-III and CUB-2011 to show the effectiveness of their proposed approaches including whether indeed adding the side-channel improves the label prediction and adding the autoregressive model improves the concept prediction accuracy.


**Questions:**

* Am I correct in understanding that the authors did not provide any result on a model that employs both side channel and autoregressive models altogether?
* Autoregressive modeling for concept prediction seems quite random and only adds for improving predictive performance. How is it relevant to addressing the leakage issue?
* It’s not clearly explained why there is a need for the path from c to z and how to train this amortization network `gamma`.


**Limitations:**

* There are a few places that appear to be written at the last minute and especially the main technical parts are not described entirely clearly.

**Strengths And Weaknesses:**

Strengths:
* The main idea and presentation of the paper is good overall.
* The experimental results on the autoregressive model (Hard AR) performing well on the concept prediction as well as the label prediction (also when intervened) are nice.
* The adaptive use of the side-channel from the amortization network shows its applicability for different scenarios.

Weaknesses:
* Perhaps it’s unfair but to me the pitch of this work (and thus the title) is a bit misleading: this paper is actually focused on closing the gap between soft and hard CBMs (in terms of the performance) rather than directly addressing the leakage issue in soft CBMs. To elaborate, adding a side channel does not prevent leakage; it helps improve predictive performance of a hard CBM which already/inherently prevents the leakage issue. If you say the leakage issue is being addressed, one would expect something like improving based upon a soft CBM, not hard one. Perhaps simply “improving hard CBMs” for example or “Addressing Two Shortcomings of Concept Bottleneck Models (the title of Section 4)” would suit better.
* Modeling correlations between concepts doesn’t quite fit the main motivation of preventing leakage; also, it’s not convincing yet why modeling correlations between concepts should be a key.
* The paper is not addressing potential limitations of the proposed approach at all, except they say it’s costly for a large data set at the end of Section 5. I think this part needs a lot more addressing on the computational overhead (for amortization network, autoregressive model, MC sampling, etc.) to make this work fair.
* There is a potential room for improvements in modeling the correlations among concepts. However, the proposed idea of using an autoregressive model seems quite under-explored.
* The two main ideas of using the side channel and autoregressive model are separate.
* The leakage is already addressed in prior work [Mahinpei et al., 2021, Margeloiu et al. 2021] and therefore the demonstration of it itself doesn’t seem to add much new insights.

---

> ### Author Response · Authors · 2022-08-02
> **[Part 1/2] Thank you for the constructive feedback on how to improve the paper.**
>
> Thank you for the constructive feedback on how to improve the paper. We address each point in our reply.
> * **[The title] is a bit misleading: this paper is actually focused on closing the gap between soft and hard CBMs (in terms of the performance) rather than directly addressing the leakage issue in soft CBMs.**
> We agree with the point that our paper does not directly fix leakage. It addresses leakage in the sense that it fixes the drawbacks of non-leaky models compared to leaky models, therefore eliminating the need to use leaky models in the first place. We are actively discussing your suggestions for an updated the title, but we have not made the final decision yet.
>
> * **Modeling correlations between concepts doesn’t quite fit the main motivation of preventing leakage; also, it’s not convincing yet why modeling correlations between concepts should be a key.  Additionally, how does autoregressive modeling for concept prediction address leakage?**
> The motivation for both of our contributions is to allow hard CBMs to model the information that soft CBMs simply leak. We identify two such information sources:
> i) Hard CBMs cannot capture correlations in concepts &rarr; we use a more flexible, autoregressive concept predictor
> ii) Hard CBMs cannot capture information not present in the concepts &rarr; we use a side channel.
>
> * **The paper is not addressing potential limitations of the proposed approach at all, except they say it’s costly for a large data set at the end of Section 5. I think this part needs a lot more addressing on the computational overhead.**
> Thank you for the suggestion. We highlight two limitations of our approach: the (possible lack of) interpretability of the side-channel and the increased training costs. We extended the computational costs section with the following table describing the training costs. Notably, the more complex image dataset, CUB-2011 takes more epochs to converge. The cost of the autoregressive architecture and the side-channel is more significant (compared to the unmodified model) when the predictive model is relatively simple, which is the case in MIMIC-III EWS.
> | Model                                   | Dataset       | Time / epoch | Training epochs |
> | --------------------------------------- | ------------- | ------------ | --------------- |
> | Soft (joint, sequential or independent) | MIMIC-III EWS | 24.2 s       | 20              |
> | Hard                                    | MIMIC-III EWS | 27.0 s       | 20              |
> | Hard AR                                 | MIMIC-III EWS | 173.4 s      | 20              |
> | Hard AR w/ side channel                 | MIMIC-III EWS | 308.7 s      | 30              |
> | Soft (joint, sequential or independent) | CUB-2011      | 52.7 s       | 60              |
> | Hard                                    | CUB-2011      | 51.0 s       | 150             |
> | Hard AR                                 | CUB-2011      | 83.2 s       | 200             |
> | Hard AR w/ side channel                 | CUB-2011      | 83.7 s       | 210             |
>
> * **There is a potential room for improvements in modeling the correlations among concepts. However, the proposed idea of using an autoregressive model seems quite under-explored.**
> Indeed, to our knowledge, we are the first ones to apply autoregressive concept predictors to CBMs, although these models are frequently used elsewhere in the literature. Future work could explore different ways of modeling the correlations among concepts.
>
> * **The two main ideas of using the side channel and autoregressive model are separate.**
> Yes, each of these separate contributions addresses a type of information that is leaked in a soft CBM. Both contributions are needed to close the gap between soft and hard CBMs.
>
> * **The leakage is already addressed in prior work [Mahinpei et al., 2021, Margeloiu et al. 2021] and therefore the demonstration of it itself doesn’t seem to add much new insights.**
> [Margeloiu et al., 2021] use posthoc interpretability methods to show that concepts learnt from CBMs are not necessarily semantically meaningful. [Mahinpei et al., 2021] discuss why leakage occurs in soft CBMs and propose that training such models should explicitly minimize the mutual information between concept dimensions if concepts are independent. Unlike these, in our paper we show leakage can be mitigated via using a side channel and autoregressive model which can enable learning more semantically meaningful concepts. While we give a description of leakage as it is the core motivation for our work, each of our experiments showcases the efficacy of our technical contributions.

---

> > ### Author Response · Authors · 2022-08-02
> > **[Part 2/2]**
> >
> > * **Am I correct in understanding that the authors did not provide any result on a model that employs both side channel and autoregressive models altogether?**
> > No, our *side channel experiments (Section 5.3) use both the autoregressive concept predictor and the side channel.* We present the results (Figure 3) this way to ensure that any observed performance disparity is due to the insufficiency of the concept set and not the inflexibility of the concept predictor.
> >
> > * **It’s not clearly explained why there is a need for the path from c to z and how to train this amortization network gamma.**
> > The amortization network ($\gamma$) is not needed in the standard setting. It is needed to i) marginalize the side channel in the scenario where the concepts must fully explain the label prediction (e.g. safety critical applications) and ii) to compute the completeness score that tells us whether the concepts are sufficient for prediction or a side-channel is needed. We use a small, two layer network as the amortization network and train it over 10 epochs. We expanded the description of the amortization network and clarified the details.
> >
> > * **There are a few places that appear to be written at the last minute and especially the main technical parts are not described entirely clearly.**
> > We took another pass and tried to clarify the technical parts, including the description of the amortization network. If you have specific areas we would be keen to get your pointers.
> >
> > We are making the following changes to the revised version to address these concerns:
> > 1. We are including the above table showing the training cost of each model.
> > 2. We are expanding the description of the amortization network.

---

### Author Response · Authors · 2022-08-02
**Revision changelog**

Thank you to all reviewers for providing constructive feedback to improve our work. We made the following changes to our manuscript using the additional page allowed for the revised version:
* Brought Appendix D into the main paper that shows that interventions remain effective in the presence of a side-channel. (Reviewer skST’s suggestion)

* Added Appendix E that presents experiments with interventions in reverse order to show that the ordering of the concepts does not affect interventions. Also added a clarification and a reference to Appendix E in Section 4.2. (Reviewer skST’s suggestion)

* Included a table detailing the computational cost of training each model. (Reviewer L4TG’s suggestion)

* Expanded the description of the amortization network. (Reviewer L4TG’s suggestion)

---

### Meta-Review · Area_Chair_bWBa · 2022-08-27

**Recommendation:** Accept
**Confidence:** Certain

**Metareview:**

The paper contains an interesting discussion on Concept Bottleneck Models pointing out their main flaws. The proposed remedies are plausible and documented empirically. The reviewers have converged to the consensus and all have scored the paper above the bar.

Minor comments:
- The caption of Figure 1 seems to be wrong. The middle part is swapped with the right part.
- It might be worth to say that the concept learning is a multi-label classification problem. The last method resembles classifier chains (particularly probabilistic classifier chains).

**Award:**

No

---

### Decision · Program_Chairs · 2022-09-14

Accept